

# Literature-based latitudinal distribution and possible range shifts of two US east coast dune grass species (*Uniola paniculata* and *Ammophila breviligulata*)

Evan B. Goldstein[1], Elsemarie V. Mullins[1], Laura J. Moore[1], Reuben G. Biel[1,*], Joseph K. Brown[2,*], Sally D. Hacker[3,*], Katya R. Jay[3,*], Rebecca S. Mostow[3,*], Peter Ruggiero[4,*] and Julie C. Zinnert[2,*]

[1] Department of Geological Sciences, University of North Carolina at Chapel Hill, Chapel Hill, NC, USA
[2] Department of Biology, Virginia Commonwealth University, Richmond, VA, USA
[3] Department of Integrative Biology, Oregon State University, Corvallis, OR, USA
[4] College of Earth, Ocean, and Atmospheric Sciences, Oregon State University, Corvallis, OR, USA
* These authors contributed equally to this work.

Corresponding author
Evan B. Goldstein,
evan.goldstein@unc.edu

## ABSTRACT

Previous work on the US Atlantic coast has generally shown that coastal foredunes are dominated by two dune grass species, *Ammophila breviligulata* (American beachgrass) and *Uniola paniculata* (sea oats). From Virginia northward, *A. breviligulata* dominates, while *U. paniculata* is the dominant grass south of Virginia. Previous work suggests that these grasses influence the shape of coastal foredunes in species-specific ways, and that they respond differently to environmental stressors; thus, it is important to know which species dominates a given dune system. The range boundaries of these two species remains unclear given the lack of comprehensive surveys. In an attempt to determine these boundaries, we conducted a literature survey of 98 studies that either stated the range limits and/or included field-based studies/observations of the two grass species. We then produced an interactive map that summarizes the locations of the surveyed papers and books. The literature review suggests that the current southern range limit for *A. breviligulata* is Cape Fear, NC, and the northern range limit for *U. paniculata* is Assateague Island, on the Maryland and Virginia border. Our data suggest a northward expansion of *U. paniculata,* possibly associated with warming trends observed near the northern range limit in Painter, VA. In contrast, the data regarding a range shift for *A. breviligulata* remain inconclusive. We also compare our literature-based map with geolocated records from the Global Biodiversity Information Facility and iNaturalist research grade crowd-sourced observations. We intend for our literature-based map to aid coastal researchers who are interested in the dynamics of these two species and the potential for their ranges to shift as a result of climate change.

## INTRODUCTION

Coastal foredunes are often the first line of protection against elevated water levels, protecting habitat and infrastructure from flooding and storm impacts (*Sallenger, 2000*). Coastal dunes are the result of ecomorphodynamic feedbacks—the presence of vegetation leads to localized sand deposition (*Arens, 1996*; *Kuriyama, Mochizuki & Nakashima, 2005*), and this burial stimulates plant growth (*Maun & Perumal, 1999*; *Gilbert & Ripley, 2010*), resulting in further sand deposition and the eventual development of a vegetated coastal foredune (*Hesp, 1989*; *Arens, 1996*; *Arens et al., 2001*; *Hesp, 2002*; *McLean & Shen, 2006*; *Zarnetske et al., 2012*; *de Vries et al., 2012*; *Durán & Moore, 2013*).

Along the northern portion of the US Atlantic coastline, *Ammophila breviligulata* Fernald (American beachgrass; perennial $C_3$ plant) is the dominant grass in dune development. In contrast, along the southern coastline, *Uniola paniculata* L. (sea oats; perennial $C_4$ plant) is the dominant dune-building grass. Other vegetation also contributes to the growth of US east coast dunes and may be locally abundant, including *Spartina patens* (saltmeadow cordgrass; *Lonard, Judd & Stalter, 2010*), *Iva imbricata* (dune-marsh elder; *Colosi & McCormick, 1978*), *Schizachyrium littorale* (shore little bluestem; *Oosting & Billings, 1942*; *Lonard & Judd, 2010*), *Carex kobomugi* (Asiatic sand sedge; *Small, 1954*; *Wootton et al., 2005*; *Burkitt & Wootton, 2011*), and *Panicum amarum* (bitter panicgrass; *Woodhouse, Seneca & Broome, 1977*; *Lonard & Judd, 2011*).

Understanding species range limits and their underlying causes has motivated more than a century of research by biogeographers and ecologists (*Grinnell, 1904*; *Mack, 1996*), and is becoming increasingly urgent for environmental management as global environmental change alters species distributions (*Parmesan & Yohe, 2003*; *Pearson & Dawson, 2003*). Descriptions of the range limits of the two dominant dune grasses of the US Atlantic coastline not only improves regional analyses of geomorphology, coastal protection services, and restoration dynamics of east coast dunes, it also provides a baseline for the study of future changes in the range limits of these important dune grasses. Morphological differences in coastal dunes of the US east coast have been attributed to a combination of factors such as forcing conditions (wind, waves, tide), dominant grain size, and vegetative controls such as dune grass species (*Godfrey, 1977*; *Godfrey & Godfrey, 1973*; *Godfrey, Leatherman & Zaremba, 1979*). For example, *Godfrey (1977)* hypothesized that *U. paniculata* and *A. breviligulata* differ in their growth rate and growth form, thereby setting the pace of dune growth as well as defining dune shape and size (i.e., hummocky dunes of *U. paniculata* vs. continuous dunes of *A. breviligulata*), an idea that is consistent with recent model results (*Goldstein, Moore & Vinent, 2017*). The effects of grass morphology and growth form on dune shape have also been shown on the US west coast, where two non-native grass species with distinct morphologies and growth characteristics produce differing dune shapes (*Hacker et al., 2012*; *Zarnetske et al., 2012*). Broadly, authors have stated that the northern range limit of *U. paniculata* is in Virginia (VA), and the southern limit for *A. breviligulata* is in North Carolina (NC), with species co-occurring in each of the states (*Duncan & Duncan, 1987*; *Silberhorn, 1999*).

Our overall objective in this study is to provide a review and synthesis of previous work on the range limit and locations of *U. paniculata* and *A. breviligulata* along the US Mid-Atlantic coast as a baseline for future investigations of possible range shifts. To achieve this, we conducted a literature search of papers that contain range limits and occurrences of one or both of the two species at or beyond the generally accepted geographic limits. Our specific goals were to: (1) determine the range boundaries through time of *U. paniculata* and *A. breviligulata* from an extensive literature survey and assess the zone of overlap between the two species; (2) investigate, through temperature trends, whether climate may be playing a role in the boundaries and potential range shifts; and (3) provide a map-based literature review (*Tobias, 2014*; *Tobias & Mandel, 2015*) to aid researchers studying the dynamics of the two grass species across their ranges and within their zone of overlap.

## MATERIALS AND METHODS

We performed a literature search on December 19th, 2017 for published studies in botany, ecology, and coastal geomorphology that specifically include four types of information, which we then collated: (1) statements regarding the northern range limit of *U. paniculata*; (2) statements regarding the southern range limit of *A. breviligulata*; (3) studies focusing on these species and their occurrences (in a coastal dune context) at the limits of the stated range, with an emphasis on examples of *A. breviligulata* in NC and southward and *U. paniculata* in VA and northward; (4) greenhouse and laboratory studies focusing on *U. paniculata* and *A. breviligulata* that may relate to their ranges (Supplemental Information 1). All co-authors participated in the search.

All relevant range data were noted in a spreadsheet shared among the co-authors along with the following information: the author designated place name (e.g., "Cape Hatteras"), the year published, citation information (e.g., book title, journal, DOI), species ("A" or "U"), if the stated species was part of an explicit planting experiment, and where in the text the comment on occurrence was made (e.g., "third column, second paragraph, page three"). Lastly, latitude and longitude were included; either those given in the text, or if not explicitly given, as estimated based on place names provided in the text.

We placed all papers that referenced *U. paniculata* and *A. breviligulata* from NC to NJ in a shared folder. We used a version of "snowball" sampling to find new publications by conducting forward and backward searches of our initial set of papers ("cited by" and "citing") in Web of Science and Google Scholar to discover new documents. We also searched for previous taxonomic names of *U. paniculata*—*Briza caroliniana* J. Lamark, *Nevroctola paniculata* C. Rafinesque-Schmaltz. ex Jackson, *Trisiola paniculata* C. Rafinesque-Schmaltz, *N. maritima* C. Rafinesque-Schmaltz ex Jackson, *U. floridana* M. Gandoger, *U. heterochroa* M. Gandoger, *U. macrostachys* M. Gandoger; sea oats (*Yates, 1966*; *Lonard, Judd & Stalter, 2011*)—and *A. breviligulata*—*A. arenaria* var. *breviligulata* (Fernald), though *A. breviligulata* has been a stable species name since the 1920s (*Maun & Baye, 1989*). Data collection was performed as a "sprint" during which time authors worked contemporaneously to assemble a database (*Goldstein et al., 2017*). We then used the "Leaflet" JavaScript library (*Agafonkin, 2017*) via an R package

**Table 1 References used in construction of the interactive map.**

| Species | Citation |
|---|---|
| A. breviligulata | Martin (1959), Harvill (1967), Woodhouse & Hanes (1967), Seneca & Cooper (1971), Singer, Lucas & Warren (1973), Schroeder, Hayden & Dolan (1979), Koske & Polson (1984), Klotz (1986), Roman & Nordstrom (1988), Conn & Day (1993), Seliskar & Huettel (1993), Seliskar (1994, 1995), Dilustro & Day (1997), Seliskar (2003), Day et al. (2004), Conaway & Wells (2005), Heyel & Day (2006), Young et al. (2011), Wolner et al. (2013), Brantley et al. (2014), Yousefi Lalimi et al. (2017) |
| U. paniculata | Lewis (1918), Hitchcock (1935), Wells (1928), Oosting & Billings (1942), Oosting (1945), Tatnall (1946), Burk (1961), Wagner (1964), Seneca (1972), Godfrey & Godfrey (1974), Harper & Seneca (1974), Stalter (1975), Cleary & Hosier (1979), Godfrey & Godfrey (1976), Godfrey, Leatherman & Zaremba (1979), Hosier & Cleary (1977), Silander & Antonovics (1982), Tyndall et al. (1986, 1987), Stallins (2002), Franks et al. (2004), Subudhi et al. (2005), Burgess et al. (2005), Zinnert et al. (2011), Hodel & Gonzales (2013), Long, Fegley & Peterson (2013a, 2013b), USDA (2013), Purvis, Gramling & Murren (2015), Mullins & Moore (2017) |
| A. breviligulata and U. paniculata | Kearney (1900, 1901); Boyce (1954) Brown (1959), Burk (1962), Woodhouse, Seneca & Cooper (1968), Seneca (1969), Godfrey & Godfrey (1973), van der Valk (1974, 1975), Levy (1976), Travis (1977), van der Valk (1977), Woodhouse, Seneca & Broome (1977), Godfrey (1977), Boulé (1979), Hosier & Eaton (1980), Hill (1986), Odum, Smith & Dolan (1987), McCaffrey & Dueser (1990), Stalter & Lamont (1990, 1997, 1999, 2000), Andrews, Gares & Colby (2002), Bachmann et al. (2002), Stallins & Parker (2003), Stallins (2005), Shafer (2010), Bright et al. (2011) |

**Note:**
A total of 82 citations; 22 for *Ammophila breviligulata*, 30 for *Uniola paniculata*, and 30 for both species.

(Cheng, Karambelkar & Xie, 2017) in R version 3.4.1 (R Core Team, 2017) to create an interactive map from the collected data (Mullins, 2018).

In addition to literature searches, we used the GBIF (The Global Biodiversity Information Facility; GBIF, 2017a) database to extract occurrence records of *U. paniculata* (GBIF, 2017b, 2018b) and *A. breviligulata* (GBIF, 2018a) on the US east coast, including data from digitized herbarium specimens and research grade iNaturalist observations (iNaturalist.org, 2018). The *U. paniculata* occurrences from GBIF contains data from queries for "*U. paniculata* L." (GBIF, 2018b) and "*U. paniculata* Roth." (GBIF, 2017b). Only GBIF records with latitude and longitude were used. This information was used as a comparison for our interactive map-based literature review.

Finally, we used long-term climate data from a NOAA meteorological station in Painter, VA, to examine annual and seasonal trends in temperature between 1956 and 2016 (Station ID: GHCND:USC00446475). This station is located near the northern range limit of *U. paniculata*. Data from 2003 is excluded due to missing observations for the month of July. Data are presented as annual mean maximum and minimum temperature and winter (Dec 21–March 20) mean temperature.

## RESULTS

In total, we found 98 unique papers/books/chapters (Tables 1 and 2) that provided 103 and 158 mentions (specific to the statements in which we searched) of *A. breviligulata* and *U. paniculata*, respectively, from 1900 to 2017 (261 total mentions; Fig. 1; Supplemental Information 2; Goldstein et al., 2017). Of the 261 total mentions in our dataset, 32 refer to range boundaries specific enough to place on a map. Of these 32 mentions spanning 1946–2013, 14 are mentions of *U. paniculata* and 18 mentions for *A. breviligulata* (Table 2; Goldstein et al., 2017). Because each mention of a range limit is tied to a citation, we were

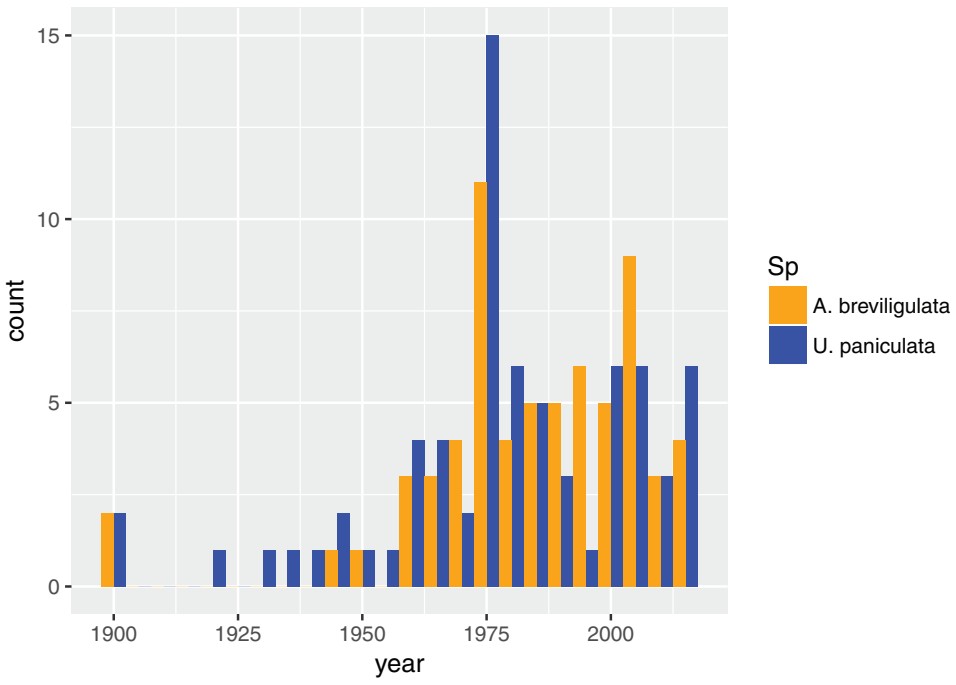

**Figure 1** **Dataset composition.** Dates for all referenced work for each species in our dataset, binned every five years.

**Table 2 References with mentions to range limits.**

| Species | Citation |
|---|---|
| *A. breviligulata* | *Brown (1959)*, *Burk (1968)*, *Godfrey & Godfrey (1973)*, *Godfrey & Godfrey (1976)*, *Godfrey, Leatherman & Zaremba (1979)*, *Maun & Baye (1989)*, *Rogers & Nash (2003)*, *Pilkey, Rice & Neal (2004)*, *Frankenberg (2012)*, *Thornhill, Suiter & Krings (2013)* |
| *U. paniculata* | *Laing (1958)*, *Wagner (1964)*, *Yates (1966)*, *Woodhouse (1982)*, *Lonard, Judd & Stalter (2011)*, *Hodel & Gonzales (2013)* |
| *A. breviligulata* and *U. paniculata* | *Hitchcock & Chase (1950)*, *Woodhouse & Hanes (1967)*, *Seneca (1972)*, *Godfrey (1977)*, *Duncan & Duncan (1987)*, *Krause (1988)*, *Overlease (1991)*, *Silberhorn (1999)* |

**Note:**
A total of 24 citations; 10 for *Ammophila breviligulata*, six for *Uniola paniculata*, and eight for both species.

able to collect temporal information on the northern range limit of *U. paniculata* and the southern range limit of *A. breviligulata* (Fig. 2). Many mentions of range limits give general geographic information, for instance limiting *U. paniculata* to the "Virginia Capes," or *A. breviligulata* to the "Outer Banks"—this geospatial imprecision prohibits a thorough regression analysis. However, the data in Fig. 2 is at least qualitatively suggestive of a slight northward trend in the stated northern range limit of *U. paniculata*. The data do not allow us to draw conclusions about temporal range shifts for *A. breviligulata*.

We compiled literature mentions of each species in geographic space by placing them on an interactive map (Fig. 3; Supplemental Information 3). The full interactive html map enables users to examine specific observations in greater detail by changing the map scale, selecting individual observations of interest, and navigating to linked primary literature via DOIs or stable URLs. The most southerly studies of *A. breviligulata* in our dataset are *Bright et al. (2011)* at Kure Beach, NC and *Hosier & Eaton (1980)*
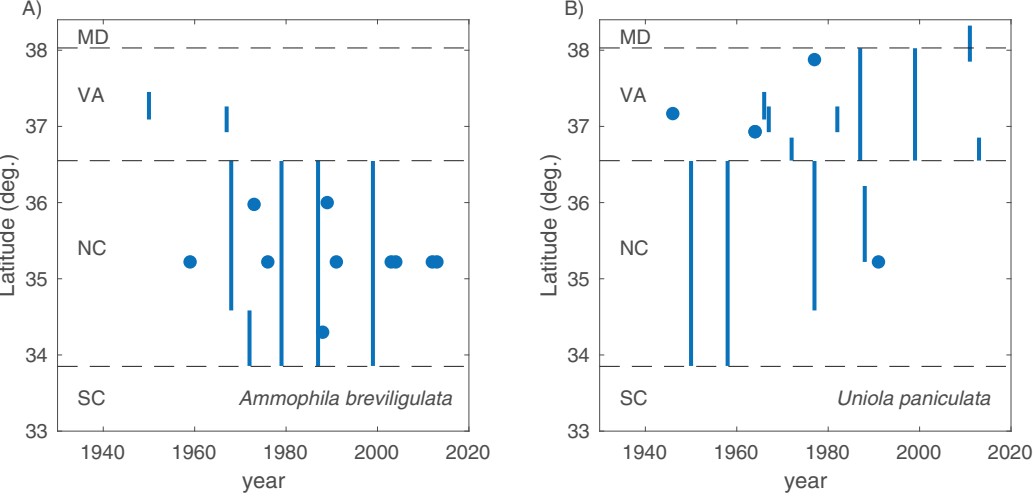

**Figure 2 Range limits.** Southern range limit for *A. breviligulata* (A) and northern range limit *U. paniculata* (B), extracted from literature sources of various age. Points are specific geographic mentions, while lines are ambiguous geographical references (e.g., "Southern North Carolina," "Virginia Capes"). Dotted lines demarcate state boundaries.

at Bald Head Beach, NC. The scarcity of references to *A. breviligulata* in southern NC stands in contrast to the many references of *A. breviligulata* farther north in NC (e.g., Bogue Banks and Cape Lookout). Our literature review suggests that *A. breviligulata* becomes sparse south of Cape Lookout, NC, with no mentions in the literature of its presence south of Cape Fear, NC.

North of the Chesapeake Bay mouth, *U. paniculata* has been observed along the uninhabited islands of the VA eastern shore (*Zinnert et al., 2011*; *Boulé 1979*; *Stalter & Lamont, 2000*; *Bachmann et al., 2002*; *McCaffrey & Dueser, 1990*; *Mullins & Moore, 2017*). Farther north, *U. paniculata* appears along Assateague Island (*Stalter & Lamont, 1990*; *Hill, 1986*; *Subudhi et al., 2005*). We can find reports of only a single stand of *U. paniculata* north of Assateague Island: in Avalon NJ, *U. paniculata* was planted by the US Department of Agriculture as a trial (*Nordstrom, 2008*). This experimental stand still exists, but reports in 2013 suggest that no natural recruitment has occurred (*USDA, 2013*). Our literature review therefore suggests that *U. paniculata* becomes sparse north of the Chesapeake Bay mouth, with only a single (human-planted) stand described north of Assateague Island, MD/VA.

To supplement this geographic analysis, our analysis of temperature trends at Painter, VA indicate a general warming trend in annual maximum (1.0 °C, $r^2 = 0.24$, $p < 0.0001$) and minimum temperature (2.0 °C, $r^2 = 0.52$, $p < 0.0001$), as well as winter minimum temperature (3.6 °C, $r^2 = 0.33$, $p < 0.0001$) since 1956 (Fig. 4).

## DISCUSSION

Thermal tolerances are often implicated in limiting the range of the two species of dune grasses (*A. breviligulata* and *U. paniculata*) considered in this study. *Godfrey (1977)* and *Lonard, Judd & Stalter (2011)* suggest wintertime temperatures limit *U. paniculata* growth

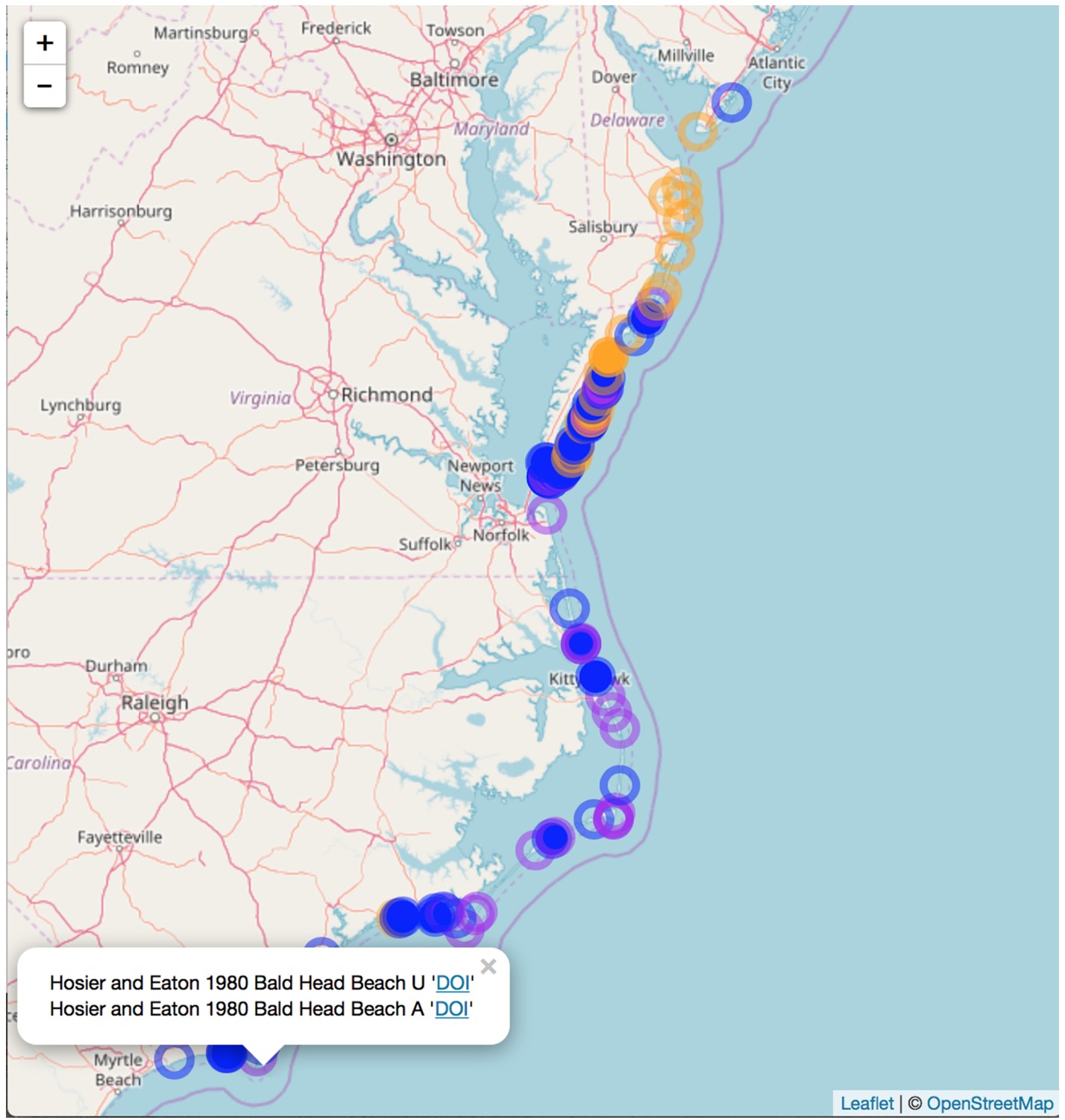

**Figure 3 Static snapshot of interactive map.** A static snapshot of the interactive map (Supplemental Information 3). The map background is OpenStreetMap data (https://www.openstreetmap.org). Each circle marker corresponds to a literature mention of a given species (orange for *A. breviligulata*, blue for *U. paniculata*, purple for both). Filled markers are literature defined locations (mentioned in the specific study). Open markers are general locations estimated by place names in the study text. In the interactive map (Supplemental Information 3), mentions can be seen within the pop-up label, as well as the corresponding species label ("A" or "U"), a location name ("Bald Head Beach") and an active link via DOI or stable URL to primary source.

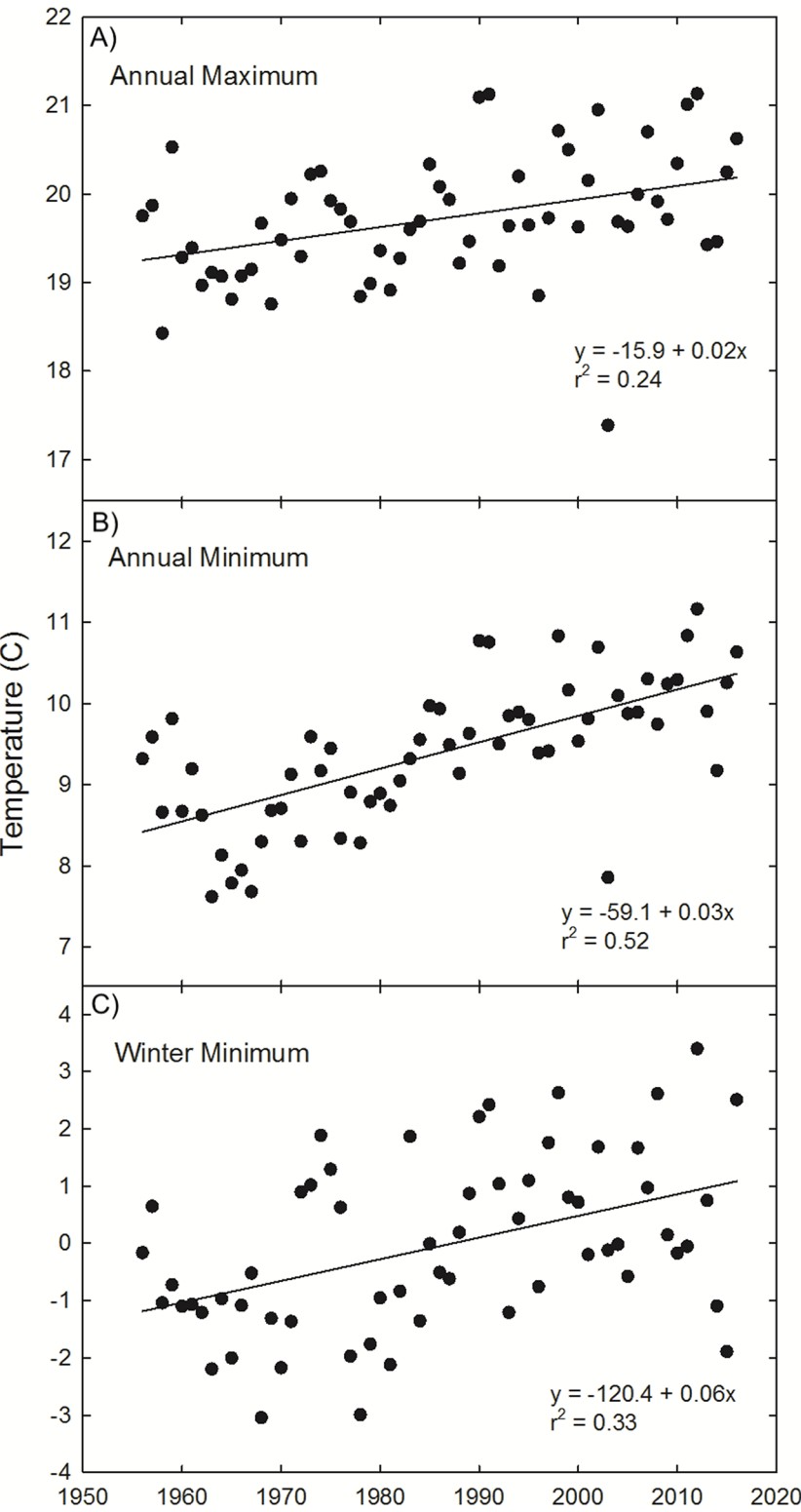

**Figure 4 Painter, VA temperature trends.** Observations and trends of increasing annual mean of the maximum temperature (A), annual mean of the minimum temperature (B), and winter (Dec 21–March 20) mean temperature (C) at Painter, VA, near the northern range limit of *U. paniculata*.

in more northern settings. *Seneca (1969, 1972)* noted that germination below 29 °C was rare for *U. paniculata* and growth was significantly reduced at low temperatures. *Westra & Loomis (1966)* and *Burgess, Blazich & Nash (2002)* also report low rates of *U. paniculata* germination with temperatures below 30 °C. Temperature analysis from Painter, VA (Fig. 4)—in the zone of overlap—indicates a winter warming trend, potentially resulting in temperatures at or near a threshold limit for successful germination and vegetative propagation of *U. paniculata* (*Westra & Loomis, 1966*; *Seneca, 1969, 1972*; *Burgess, Blazich & Nash, 2002*). Although there are few observations of *U. paniculata* along the Virginia barrier islands, populations planted experimentally in 2013 have thrived (E. de Vries et al., 2018, unpublished data). Experimentally planted *U. paniculata* in the higher latitudes of NJ show no natural recruitment (*USDA, 2013*).

We found no indication of temporal trends in the stated range limit for *A. breviligulata* in the literature. However, early studies indicate a scarcity of *A. breviligulata* in southern NC before a history of extensive plantings. *Lewis (1918)* remarks on the lack of availability of *A. breviligulata* in Beaufort, NC for planting "barrier dunes"—suggesting instead the use of *U. paniculata*. *A. breviligulata* is also missing from a Bogue Banks survey by *Burk (1962)*. In contrast to *Lewis (1918)*, *van der Valk (1975)* notes that the NC Outer Banks were planted with *A. breviligulata* instead of *U. paniculata* during campaigns in the 1930s and 1950s. *Schroeder, Dolan & Hayden (1977)* and *Godfrey (1977)* also mention plantings of *A. breviligulata* along the NC coastline and Outer Banks. *Godfrey (1977)*, *Travis (1977)*, as well as *Maun & Baye (1989)* note that *A. breviligulata* plantings occur beyond the probable "natural" range (i.e., too far south). *Seneca & Cooper (1971)* find reductions in *A. breviligulata* biomass as temperatures exceed 26°C. In addition to thermal constraints, *Woodhouse, Seneca & Broome, (1977)* and *Singer, Lucas & Warren (1973)* discuss pest and disease pressure in southern populations of *A. breviligulata*, as does *Seliskar & Huettel (1993)* for mid-Atlantic *A. breviligulata* populations.

Several studies that are not included in the map (because they describe greenhouse experiments) are relevant to understanding shifting range limits of these species and interactions that contribute to present-day range limits. These recent experiments focused on species interactions between *A. breviligulata* and *U. paniculata* (*Harris, Zinnert & Young, 2017*; *Brown, Zinnert & Young, 2017*), which are likely to be most important in their zone of overlap from NC to VA. *Harris, Zinnert & Young (2017)* found that *U. paniculata* reduces growth of *A. breviligulata* by altering physiological performance at temperatures consistent with summertime on the Virginia barrier islands. *Brown, Zinnert & Young (2017)* expanded these results by showing that leaf elongation and root length of *A. breviligulata* are reduced through interactions with *U. paniculata*. This reduction in performance may explain the observations of *A. breviligulata* plantings being displaced within 6–10 years by native *U. paniculata* along Core Banks, NC (*Woodhouse, Seneca & Cooper, 1968*).

The dominant dune grass species in a given area influences the protective services of coastal dunes. *Woodhouse, Seneca & Broome (1977)* notes that *A. breviligulata* tends to grow faster than *U. paniculata* and spread more rapidly after transplant growth.
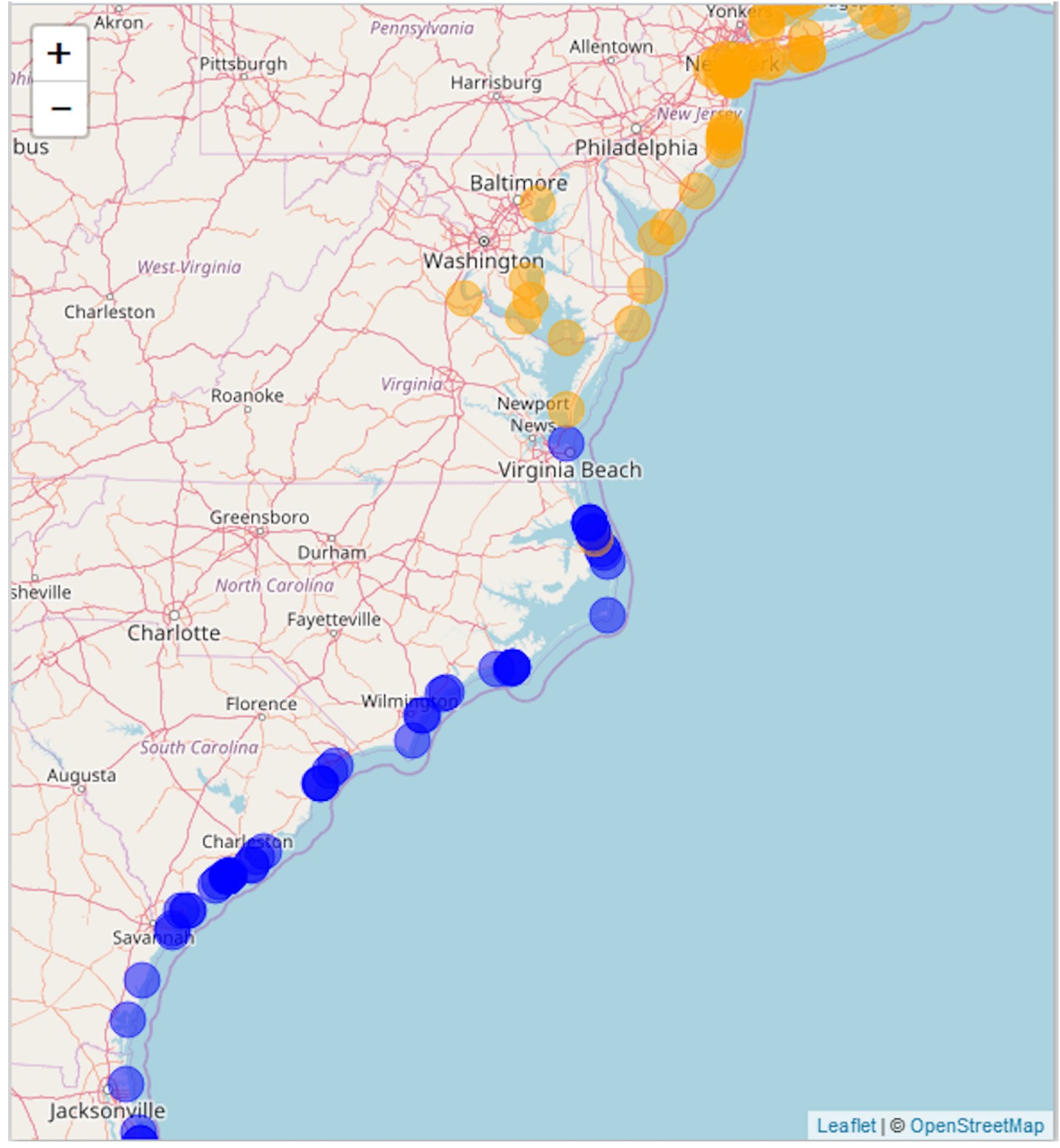

**Figure 5  GBIF data.** Map of *A. breviligulata* (orange) and *U. paniculata* (blue) occurrences from the GBIF database. The map background is OpenStreetMap data (https://www.openstreetmap.org).                                   
These differences in plant growth rate have implications for dune morphology, which have been observed in the field (*Woodhouse, Seneca & Broome, 1977*) and explored in numerical models of coastal dune growth (*Goldstein, Moore & Vinent, 2017*). These studies suggest that dunes dominated by *A. breviligulata* coalesce faster than those formed by *U. paniculata*, resulting in high, continuous dune ridges compared to hummocky dune formations associated with *U. paniculata*. Further exploration of species interactions in the zone of overlap is needed to fully understand the implications of potential changes in species composition for dune building under future climate change.

Although the focus of our study is on cataloging and mapping data from the literature, absences of *A. breviligulata* or *U. paniculata* in particular areas are also worth noting. For example, we found no reference to *A. breviligulata* south of Cape Fear, NC. However, there are suggestions in the literature that *A. breviligulata* has been planted further south. *Woodhouse & Hanes (1967)* advise that *A. breviligulata* can survive when planted for dune restoration purposes as far south as the South Carolina border with North Carolina. *Maun & Baye (1989)* discuss the presence of planted, ephemeral populations in South Carolina, Georgia, and Florida, but only cite personal communications (with E.D. Seneca) and provide no specific locations. However, comprehensive studies by *Stalter (1974*, *1975)* did not mention the occurrence of *A. breviligulata* at several sites in coastal SC. Paired surveys by *Stallins (2002*, *2005)* and *Stallins & Parker (2003)* also do not record the presence of *A. breviligulata* in Georgia locations (as compared to NC sites in the same study). This evidence suggests that *A. breviligulata* has, at times, been planted or found south of Cape Fear, NC but fails to persist.

We compare our literature-derived results (Fig. 3) to the GBIF dataset extracted for this study (*GBIF, 2017a*, *2017b*, *2018a*, *2018b*). Mapping GBIF data associated with known latitude and longitude (636 points) leads to a zone of overlap from Kitty Hawk, NC (southernmost observation of *A. breviligulata*) to Cape Henry, VA (northernmost observation of *U. paniculata*; Fig. 5). Our literature-derived results yield a larger zone of overlap (from Cape Fear, NC to southern NJ), and records many observations from within the zone of overlap (Fig. 3).

## CONCLUSION

Our literature review suggest the current southern range limit for *A. breviligulata* is Cape Fear, NC, and the northern range limit for *U. paniculata* is Assateague Island, at the border of Maryland and Virginia. The ranges for these two species overlap between Virginia and North Carolina. Results suggest a northward expansion of *U. paniculata*, possibly associated with warming trends, while the data for *A. breviligulata* remain inconclusive.

We acknowledge that there may be additional information on the ranges of these two dune grass species in theses and local guides (*Denslow, Palmer & Murrell, 2010*). These sources—as well as scanned herbarium sheets from museum collections—constitute "dark data," data not discoverable because of problems in indexing, storage, and retrieval (*Heidorn, 2008*). For this reason, a more complete picture of range limits and species

abundances should come from contemporaneous, modern, synoptic field surveys of *U. paniculata* and *A. breviligulata* throughout the zone of overlap—from NC to NJ. Given the interest in dunes as a means for providing storm protection, it would also be useful to explore the geographic variation of the vigor and survival of natural versus planted stands of these two grasses, including the effects of species interactions.

## ACKNOWLEDGEMENTS

We thank Bianca Charbonneau, Louise Wootton, Michele Tobias and the Editor for reviews that improved the manuscript. Map data copyrighted OpenStreetMap contributors and available from https://www.openstreetmap.org.

### Funding

This work was supported by NOAA (EESLR NA15NOS4780172), NSF-GLD (EAR-1324973), and the Virginia Coast Reserve Long-Term Ecological Research Program (NSF DEB-123773). Support for Elsemarie Mullins was also provided by the NSF GRFP (DGE-1650116). The funders had no role in study design, data collection and analysis, decision to publish, or preparation of the manuscript.

### Grant Disclosures

The following grant information was disclosed by the authors:
NOAA: EESLR NA15NOS4780172.
NSF-GLD: EAR-1324973.
Virginia Coast Reserve Long-Term Ecological Research Program: NSF DEB-123773.
NSF GRFP: DGE-1650116.

### Competing Interests

The authors declare that they have no competing interests.

### Author Contributions

- Evan B. Goldstein conceived and designed the experiments, performed the experiments, analyzed the data, prepared figures and/or tables, authored or reviewed drafts of the paper, approved the final draft.
- Elsemarie V. Mullins conceived and designed the experiments, performed the experiments, analyzed the data, prepared figures and/or tables, authored or reviewed drafts of the paper, approved the final draft.
- Laura J. Moore conceived and designed the experiments, performed the experiments, analyzed the data, authored or reviewed drafts of the paper, approved the final draft.
- Reuben G. Biel performed the experiments, analyzed the data, authored or reviewed drafts of the paper, approved the final draft.
- Joseph K. Brown performed the experiments, analyzed the data, authored or reviewed drafts of the paper, approved the final draft.

- Sally D. Hacker performed the experiments, analyzed the data, authored or reviewed drafts of the paper, approved the final draft.
- Katya R. Jay performed the experiments, analyzed the data, authored or reviewed drafts of the paper, approved the final draft.
- Rebecca S. Mostow performed the experiments, analyzed the data, prepared figures and/or tables, authored or reviewed drafts of the paper, approved the final draft.
- Peter Ruggiero performed the experiments, analyzed the data, authored or reviewed drafts of the paper, approved the final draft.
- Julie C. Zinnert performed the experiments, analyzed the data, prepared figures and/or tables, authored or reviewed drafts of the paper, approved the final draft.

## Data Availability

Goldstein E., E. Mullins, L. Moore, R. Biel, J. Brown, S. Hacker, K. Jay, R. Mostow, P. Ruggiero, J. Zinnert. 2017. Locations of published studies focused on Uniola paniculata and Ammophila breviligulata along the US east coast (NJ-NC). Environmental Data Initiative http://doi.org/10.6073/pasta/bdbe9a609e0508fdb7e39bc41f75bf6f.

E. Mullins. 2018. elsemar/EastCoastDuneGrass: Map for Lit review http://doi.org/10.5281/zenodo.1228461.

## Supplemental Information

Supplemental information for this article can be found online at http://dx.doi.org/10.7717/peerj.4932#supplemental-information.

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
