# Peer review of "Literature-based latitudinal distribution and possible range shifts of two US east coast dune grass species (Uniola paniculata and Ammophila breviligulata)"

_PeerJ, doi:10.7717/peerj.4932_

## Round 0.1 · original submission · Minor Revisions

The reviewers have provided suggestions apart from minor grammatical improvements. For example, Reviewer 1 notes that:

1) Parts of the discussion (lines 153-159) belong in methods and results as opposed to the discussion and then similarly, the discussion of the topic or warming and germination (line 160) needs to be fleshed out more in the discussion. I have included references for the authors in the comments of my PDF.

Reviewer 2 notes that:

Figure 5 (GBIF data) where the large Symbol size may be obscuring data points. That or there’s only one Ammophila observation in this entire data set within the zone of overlap. Given that this image isn’t zoomable, and there doesn’t seem to be a table of these data even in the Supplemental materials from which one could check actual observations, an insert focusing on the zone of overlap with more detail on the records of each species in that area may be beneficial. Either that or a supplementary data table similar to that provided for the literature-derived data sets might be provided to allow detailed follow up of the locations for each sample documented (Actually I think this might be a valuable addition to the paper, even if the figure is modified to allow more detailed display of the data in the zone of overlap

In conclusion, the manuscript with minor revision has been accepted for publication in PeerJ.

·

Basic reporting

The authors use clear writing throughout and did an excellent job researching their topic. There were a few small grammatical errors that I noted in my PDF. The three goals laid out in the intro end were great and all topics explored by the authors in great detail.

With regards to the figures:
1) please denote the number of citations found in total as well as per group in the figure captions for table 1 and 2.
2) Please change figure organization for table 1 and 2 to be chronological as opposed to alphabetical.
3) Figure 2 - having a simple line map from lat 31 to 39 shown (overlayed in some way or perhaps above both graphs?) would really help strengthen this figure.

With regards to structure:
1) Parts of the discussion (lines 153-159) belong in methods and results as opposed to the discussion and then similarly, the discussion of the topic or warming and germination (line 160) needs to be fleshed out more in the discussion. I have included references for the authors in the comments of my PDF.
2) I believe that a lot of the discussion actually belongs in the introduction. While the introduction has a lot of information, I believe that it is too short and topic specific as opposed to more broadly specific on range shifts in general- ie other non-coastal plants that have undergone range shifts and even animals would all be great topics for the authors to discuss that would greatly improve the broad interest level and applicability of the paper in the intro.
3) in general, I believe that the discussion should be rewritten to put the author's findings in the context with what has been found previously. As it currently reads, I felt the authors very much understood what their findings show, but struggled to show not how this fits into the realm of other findings and the applied context for ecosystem services, which is a topic the authors indicate is important in the introduction. More clear topic sentences laying out the broad topic of the paragraph in terms of the finding to be discussed would help make it feel less like information is just being piled on instead of put into context.


Of note:
1) 260 mentioned throughout as the sample size, but spread sheet shows 264?
2) Interactive map link either currently inactive or does not work as I could not review this.
3) Unclear what supplementary files were what as they are labelled in the document and in the reviewer package differently
4) The USDA stated range for both of these species differs from that of the authors in that it is much larger. I believe that the authors should reference and discuss this given that this is a very USA centered paper. Below are the links to this information:
-Ammophila breviligulata - https://plants.usda.gov/core/profile?symbol=AMBR
-Uniola paniculata - https://plants.usda.gov/core/profile?symbol=UNPA

Experimental design

Great! Seems like a very solid literary review of the searchable published literature. I have one issue. On line 86, the authors state that

"E.B. Goldstein and E.V. Mullins wrote the search protocol with guidance from L.J. Moore to determine inclusion/exclusion criteria. All the authors participated in the search."

I believe that this aspect of the research is important as knowing it would allow someone else to use their search criteria for a similar review style study. However, no more detail is given beyond this though it should be more clear. Similarly, this statement is strange to me as it denotes a serious responsibility to two authors Mullins and Goldstein and vague responsibility to all else who assumedly "participated in the search." Given that the search was only one day, on December 19, 2017 (as stated) and participation in this would not in most cases result in a publication for all those that used a search engine...I think this line needs to be removed and all responsibilities discussed for all involved or none singled out as such.

Validity of the findings

The findings are important, interesting, and relevant. The science is sound. As stated previously, I think the authors need to rework their intro and discussion. The into in my opinion need to be more specific to range shifts in general, not just of 'their' species and also the implications of the range shift with regards to ecosystem services. The discussion doesn't discuss their findings as much as it does others, but they are not necessarily in context with one another though they should be.

Additional comments

I made my review in a PDF editor on a tablet so there are both hand written notes in and typed PDF-style notes - I hope that you find them legible where hand-written! I colored the line numbers green where I made a note, change, addition, etc, so that they would be easy to find and not miss. I included a number of citations in the notes that should be helpful to the authors. I think that given that this paper is being submitted to an open journal, the authors would like to see it viewed by a broader audience, otherwise why would they be paying the publication fee when JCR would take this and it would reach the coastal audience. Given this, I strongly feel that the authors should make this paper more broadly scoped as opposed to jumping immediately into coastal dunes. Similarly, this is a topic very interesting and relevant to climate change, but climate change is a very peripheral topic as discussed throughout with the depth.

·

Basic reporting

I found this paper to be largely well and clearly written, with a strong review of the context for the study and an unusually clear statement of the goals of the study. The article structure was sound with easy to follow, well constructed figures and tables, and access to most of the raw data needed to support the study.

I have a few minor concerns that are really more requests for increased clarity:

Figure 5 (GBIF data) where the large Symbol size may be obscuring data points. That or there’s only one Ammophila observation in this entire data set within the zone of overlap. Given that this image isn’t zoomable, and there doesn’t seem to be a table of these data even in the Supplemental materials from which one could check actual observations, an insert focusing on the zone of overlap with more detail on the records of each species in that area may be beneficial. Either that or a supplementary data table similar to that provided for the literature-derived data sets might be provided to allow detailed follow up of the locations for each sample documented (Actually I think this might be a valuable addition to the paper, even if the figure is modified to allow more detailed display of the data in the zone of overlap)

Discussion 175 Needs clarification “Seneca (1969, 1972) found that A. breviligulata had higher germination rates at low temperatures of 18˚C.” Winter low? Spring low? Summer low? Higher than what? Low temperatures below this? Low temperatures higher than this?

Discussion 188. Again needs clarification: “This reduction in performance may explain the lack of a clear southern range shift”. Why would one expect a Southern range shift in Ammophila? Rather to the contrary, I’d expect to see this species’ range shifting north as climate warms. The data described on Uniola’s abilty to outcompete Ammophila in areas where the two co-exist would tend to exacerbate this, I would have thought. Maybe I’m misunderstanding something here but ….

Discussion 190-191 There is a logical dissonance for me in following a series of statements about how Uniola outcompetes Ammophila where the two species coexist with the statement “Woodhouse et al. (1977) notes that A. breviligulata tends to grow faster than U. paniculata and spread more rapidly.” It seems to me that a modifying phrase is needed here to the effect of “in areas where each is growing absent the presence of the other, A. breviligulata tends to grow faster than U. paniculata and spread more rapidly” or something similar.

Paragraph starting Discussion 199: I am not clear what the Author’s point is in this paragraph. They state that observations of Ammophila are absent from more southerly locations despite being planted there. Is this because the dataset is incomplete and Ammophila would be found there if a more concerted search was conducted? Or is it that Ammophila may have been planted in these locations but failed to thrive due to heat / competition with Uniola and therefore this species is no longer present in these locations?

Experimental design

The Methods were clear and well documented. The meta-analysis was appropriately conducted and appears to have been a rigorous attempt to collect all relevant data, including use of older Genus species names.

The PRISMA flow diagram for literature review was helpful in visualizing the process and would make a strong addition to the methods section of the paper. If space precludes its inclusion in the paper itself, I’d at least like to see its inclusion in the supplemental materials with a specific reference to this flow chart from the materials and methods.

Validity of the findings

The results documented Figures 2 and 3 represent valuable contributions to the field. The interactive map is particularly nicely collated (and rather fun to explore!). The authors are very clear on what they can quantify (eg trends in temperature data) and what is only a qualitative trend (eg data on changes in Northern Range of Uniola). The conclusions drawn from the data are appropriate and well substantiated.

Additional comments

I found a number of s
maller errors especially in the references cited format that probably should be fixed before publication. My apologies for the knit-picking, but I'm currently teaching a class with a focus on citation and plagiarism, so my radar is on high for these at this point!

Line 50 introduction. Need to Fix spelling of "Wootton et al."

207-8 Discussion I think there is an extra word in this sentence “also did not mention the occurrence of A. breviligulata IN AT several sites in coastal SC".

216 Discussion …"observations from within the zone of overlap (Figure 4)" references the figure number incorrectly. Figure 4 is the figure about temperature change. It is not a map of zones of overlap


References

525 Incomplete citation

507 Missing Journal name

292, 370, 390 , 412, 437, 479, 480, 486, 537 Missing volume number

253 342, 346, 381, 422, 491, 503, 505, 523, 530, 532 Italicize journal name

261, 278, 280, 282, 294, 296, 299, 344, 357, 359, 365, 368, 372, 374, 378 , 384, 397, 400, 402, 410 , 422, 424, 427, 436, 440, 447, 450, 452, 456, 458, 460, 462, 467, 470, 473, 474, 477, 482, 484, 488, 492, 503, 505, 507, 512, 514, 516, 518, 520, 530, 539 comma not colon between issue and pages

284. 291, 383, 389, 391, 392, 395, 405, 415, 447, 449, 451, 454, 460, 462, 470, 471, 490, 515, 516, 517, 534, 536 Italicize genus species

301 engineering should be capitalized

326 east coast should be capitalized

348 trends should be capitalized

312, 358, 364, 375, 383, 385, 423, 430, 439 , 454, 457, 459, 496, 511, 517, 531 Words in title should be lower case except proper nouns

351 Punctuation of initials in Ed names: should not be periods

352 spurious dash in middle of Soci-ety, Journal name should be italicized

354 spurious dash in the middle of geomor -phology Journal name should be italicized

544 spurious dash in middle of Ecol-ogy, Journal name should be italicized

368, 445, 473, 475, 492 background highlighting of DOI number

386, 527. Remove All Caps

403. Capitalize proper noun “Island Beach State Park”

487, 493 Remove comma between name and initial (Stalter, R) (Tatnall, RR)

·

Basic reporting

1. The abstract needs more detail in describing the methods. On line 26, it refers to “other records”. I think listing your other data sources (in particular iNaturalist) would not only make the abstract a better summary of the paper, but might drive more interest in the work itself.
2. The abstract should also briefly describe all the methods used such as the temperature analysis and the herbarium and iNaturalist records. I was surprised to see these in the paper since it wasn’t mentioned in the abstract. It’s good work, so let people know it’s in there.
3. The Acknowledgements section contains funding information that should get moved to the Funding Statement.

Experimental design

1. The Materials & Methods section needs to be expanded to include all of the analysis discussed in the Results and Discussions sections. The Discussion section talks about the thermal tolerances of the grasses in the first paragraph, but the methods of investigation aren’t in the Methods section.
2. The Discussion section contains results of the thermal range investigation that should be moved to the Results section (around lines 155-159).

Validity of the findings

no comment

Additional comments

The work you’ve described in this paper shows a novel, interesting, and useful synthesis of several sources of data. The integration of non-traditional data like the published scientific literature and herbarium records is something that interests me a great deal and I’m glad to see your take on this.

The paper is clearly written and easy to understand.

---

## Round 0.2 · accepted · Accept

The manuscript is improved and ready for publication. Thank you for thoroughly addressing the reviewers comments.

# ·

Basic reporting

Looks great. Very clear and understandable. References sufficient and relevant and figures that were already god are now even better

Experimental design

experimental design was always sound. Changes made to the presenting of it in the methods have greatly improved the clarity.

Validity of the findings

solid

Additional comments

Great job addressing the changes requested by all reviewers.

·

Basic reporting

no comment

Experimental design

no comment

Validity of the findings

no comment

Additional comments

This paper is much improved. The methods description is much more clear. Thank you for taking the time to incorporate the changes suggested by all the reviewers.